# Useful MRI Findings for Minimally Invasive Surgery for Early Cervical Cancer

**DOI:** 10.3390/cancers13164078

**Published:** 2021-08-13

**Authors:** Byung Kwan Park, Tae-Joong Kim

**Affiliations:** 1Department of Radiology, Sungkyunkwan University School of Medicine, Samsung Medical Center, Seoul 06351, Korea; 2Department of Obstetrics & Gynecology, Sungkyunkwan University School of Medicine, Samsung Medical Center, Seoul 06351, Korea

**Keywords:** early cervical cancer, magnetic resonance imaging, minimally invasive surgery

## Abstract

**Simple Summary:**

Radical hysterectomy and lymph node dissection are extensive procedures with severe post-operative morbidities and should be avoided on patients with low risk of recurrence. Still, due to lack of good prognostic tools, radical surgery is performed on most patients with early stage cervical cancer, leading to overtreatment and unnecessary morbidities. The recent International Federation of Gynecology and Obstetrics (FIGO) staging system accepts the use of magnetic resonance imaging (MRI) in addition to physical examination. Currently, 3 Tesla (3T) MRI is available widely and, due to its high soft tissue contrast, can provide more useful information on precise estimation of tumor size and metastasis than can physical examination in patients with cervical cancer. Therefore, this imaging modality can help gynecologic oncologists to determine whether minimally invasive surgery is necessary and can be used for early detection of small recurrent cancers.

**Abstract:**

According to the recent International Federation of Gynecology and Obstetrics (FIGO) staging system, Stage III cervical cancer indicates pelvic or paraaortic lymph node metastasis. Accordingly, the new FIGO stage accepts imaging modalities, such as MRI, as part of the FIGO 2018 updated staging. Magnetic resonance imaging (MRI) is the best imaging modality to estimate the size or volume of uterine cancer because of its excellent soft tissue contrast. As a result, MRI is being used increasingly to determine treatment options and follow-up for cervical cancer patients. Increasing availability of cancer screening and vaccination have improved early detection of cervical cancer. However, the incidence of early cervical cancers has increased compared to that of advanced cervical cancer. A few studies have investigated if MRI findings are useful in management of early cervical cancer. MRI can precisely predict tumor burden, allowing conization, trachelectomy, and simple hysterectomy to be considered as minimally invasive treatment options for early cervical cancer. This imaging modality also can be used to determine whether there is recurrent cancer following minimally invasive treatments. The purpose of this review is to highlight useful MRI features for managing women with early cervical cancer.

## 1. Introduction

More than 600,000 women are diagnosed with cervical cancer annually and the disease causes over 300,000 deaths worldwide [1]. However, due to the establishment of worldwide screening programs leading to early detection of disease, an increasing portion of newly detected cases are early stage. Minimally invasive surgery is becoming an increasingly important treatment strategy.

Radical hysterectomy and lymph node dissection are standard treatment options for IB1 cervical cancer [2,3,4,5]. Several investigators have reported that this treatment can improve long-term survival [2,3,4,5]. However, many women suffer from various post-surgical morbidities, such as bladder dysfunction [4,6,7], sexual dysfunction [8,9], anorectal motility disorders [8,9], and lymphedema [10,11,12]. Their quality of life can be affected negatively by aggressive or excessive treatment despite the small size of early cervical cancer. Minimally invasive treatment is necessary to reduce such functional disability and is defined as all surgical treatments without using a term “radical”.

Magnetic resonance imaging (MRI) is more precise at estimating tumor volume than is physical examination because it offers good soft tissue contrast, which enables accurate measurement of three-dimensional axes [13,14,15,16,17,18]. Recent FIGO staging guidelines are based on lymph node metastasis in the abdomen and pelvis [19,20,21,22,23]. However, it is not possible to assess lymph node metastasis by physical examination alone [24,25]. Therefore, computed tomography (CT) or MRI is necessary to detect metastatic lymph nodes. In contrast to CT, MRI and positron emission tomography (PET) scan can provide anatomical and functional information to detect metastatic lymph nodes [13,26,27,28].

As a result, MRI can play an important role in determining whether minimally invasive surgery is necessary in patients with early cervical cancer [14,21,22,29]. This imaging examination is also useful for assessing recurrent cervical cancer following conization or trachelectomy in women who want to preserve fertility. Few studies have investigated pre-operative or follow-up MRI findings and how they impact minimally invasive management of cervical cancer. The definition of early cervical cancer is controversial. We define early cervical cancer as FIGO stage IA and IB cervical cancers for which robot-assisted surgery provides reasonable treatment outcomes [30]. Our purpose in this review is to highlight useful MRI features for managing women with early cervical cancer before and after treatment. Another purpose of our review is to help radiologists know how to make good patient preparations, to make good gynecologic MR images at the 3T scanner, and what imaging features are important for surgical plan.

## 2. MRI Protocols

Table 1 shows the parameters of pelvic MRI at Samsung Medical Center. The MRI scanner is an Ingenia CX scanner (Philips Healthcare, Best, The Netherlands), and a phase-array multicoil (Philips Healthcare) is used. After the pelvis MRI scan is finished, the upper abdomen can be scanned by moving the surface coil upward.

### 2.1. Patient Preparation and Technical Tips

3T MRI is being used increasingly in practice because it provides faster scanning and higher resolution than 1.5T MRI [31,32,33]. Increased magnetic strength improves signal-to-noise ratio but is associated with a high frequency of MRI artifacts. Technical tips to achieve good MRI quality include reducing bowel movement, using a high matrix size, setting a high field of view, and achieving a reasonable scan time. An anti-peristaltic agent should be injected intramuscularly or intravenously 20 min before MRI scan [34]. Bowel peristalsis should be inhibited because bowel movement disturbs the magnetic field [35]. Moreover, we ask patients to fast for 6 h to reduce bowel movement, although this is controversial.

Matrix size should be increased to improve image resolution [36], and the field of view should be increased to reduce the presence of aliasing artifacts [37]. However, a larger matrix size results in decreased signal-to-noise ratio and a longer scan time [36]. Therefore, the optimal matrix size should be determined in each institute, and the scan time should be approximately 5 min per MRI sequence. A long scan time is more likely to be accompanied by patient movement than is a short scan, which can degrade image quality although image resolution is improved. Every institute needs to determine the appropriate matrix sizes of pelvic MRI sequences by means of trial and error. Optimal matrix sizes cannot be predetermined because they vary according to MRI scanner. It took us approximately one year to determine the optimal matrix sizes, and we continue to modify them every time MRI scanners are changed or MRI software is upgraded.

### 2.2. T2-Weighted Imaging (T2WI)

The scan range of T2WI should be modified to completely cover the uterus, vagina, vulva, and adnexa. Therefore, a localization scan prior to T2WI is essential to determine the anatomy of these organs. T2WI should be performed in the axial, sagittal, and coronal planes [21,38,39] (Figure 1) (Table 1). The axial scan axis is perpendicular to the uterine cervix. The sagittal or coronal scan axis is parallel to the uterine cervix. The repetition time of axial T2WI at our institute is 4044 ms, whereas that of sagittal or coronal T2WI is 3000 ms. The field of view of axial T2WI is 28 × 28 cm, and that of sagittal or coronal T2WI is 30 × 30 cm. All T2WI scans performed at our institute have the following parameters (Table 1): echo time, 100 ms; matrix size, 640 × 640; slice thickness, 4 mm; interslice spacing, 4.4 mm; echo train length, 16; number of excitations, 1; flip angle, 90°.

### 2.3. T1-Weighted Imaging (T1WI)

We obtain T1W images using a fast spin echo (Table 1). The scan plane is obtained in the axial axis. The scan range of axial T1WI is similar to that of axial T2WI. The axial scan axis is perpendicular to the uterine cervix. We use the following MR parameters for T1WI scans (Table 1): repetition time, 500 ms; echo time, 10 ms; matrix size, 384 × 381; slice thickness, 5 mm; interslice spacing, 7 mm; field of view, 28 × 28 cm; echo train length, 4; number of excitations, 1; flip angle, 90°.

### 2.4. Diffusion-Weighted Imaging (DWI)

DWI is performed using echo planar imaging. The scan axis is perpendicular to the cervix and is obtained in the axial plane. This sequence is useful to differentiate cervical tumors from normal tissue or to detect lymph node metastasis [21,22] (Figure 1). The scan range of axial DWI is similar to that of axial T2WI or T1WI. DWI has the following MR parameters (Table 1): repetition time, 6900 ms; echo time, 64 ms; matrix size, 144 × 206; slice thickness, 4 mm; interslice spacing, 4.4 mm; field of view, 28 × 28 cm; echo train length, 83; number of excitations, 3; b values, 0, 100, 1000 s/mm^2^; flip angle, 90°. Apparent diffusion coefficient (ADC) map images are created using all b values (0–100–1000 s/mm^2^) after DW images are obtained.

### 2.5. Contrast-Enhanced Imaging (CEI)

Dynamic contrast-enhanced images are obtained at our institute using a 3D T1 mDIXON. The axial scan axis is perpendicular to the uterine cervix. The scan range of axial dynamic CEI is smaller than that of axial T2WI or T1WI. The axial scan axis is perpendicular to the uterine cervix. Dynamic CEI scans are acquired using the following MR parameters (Table 1): repetition time, 3.5 ms; echo time, 0 ms; matrix size, 208 × 207; slice thickness, 4 mm; interslice spacing, 2 mm; field of view, 28 × 24 cm; echo train length, 2; number of excitations, 1; flip angle, 10°. Unenhanced images are scanned, and dynamic CE images are captured at 1, 2, and 3 min after contrast material is injected intravenously.

After dynamic CEI, delayed CEI is performed using a fast spin echo sequence (Figure 2). The scan range and scan axis are similar to those of sagittal T2WI. This sequence has the following MR parameters (Table 1): repetition time, 596 ms; echo time, 10 ms; matrix size, 384 × 380; slice thickness, 4 mm; interslice spacing, 4.4 mm; field of view, 30 × 30 cm; echo train length, 5; number of excitations, 1; flip angle, 90°.

## 3. MRI Findings

### 3.1. T2WI Findings

T2WI is the most basic and essential MRI sequence and can depict details of the uterine anatomy. Early cervical cancer is homogeneously hyperintense on T2WI compared to cervical stroma [13,27,40]. As the cystic component increases, tumor signal intensity becomes heterogeneously high. This MR sequence can provide precise estimation of cervical cancer volume [32,33,38]. Therefore, small cervical cancer is visible on T2WI if the tumor size is larger than 5 mm [14,29]. However, it is not uncommon for early cervical cancer to be invisible on T2WI because a small tumor can be removed by biopsy alone [14,29] (Figure 1 and Figure 2). In other words, invisible cancer on T2WI strongly suggests a very small-volume cervical tumor. Accordingly, even though cervical cancer is stage IB1 based on physical examination, radical parametrectomy is not necessary if the tumor is not visible on MRI (Figure 1). Roh et al. reported that an MRI-invisible diagnosis in early-stage cervical cancer patients was associated with a high probability of a false negative result and was associated with underdiagnosis [41]. Invisible cancer on MRI does not indicate absent cancer but rather a small tumor volume [14]. Among patients with invisible cancers on MRI, approximately 50% have no residual tumor, and the other 50% have small residual tumors ranging in size from 4–5 mm [14,41]. Accordingly, minimally invasive surgery is indicated for cervical cancers that are invisible on MRI.

### 3.2. T1WI Findings

Cervical cancer is iso-intense on T1WI compared to cervical stroma. Therefore, this MR sequence is not ideal for detection of early cervical cancer because of poor tumor-to-stromal contrast. However, this sequence is useful to differentiate hemorrhage and fat because these are hyperintense on T1WI. Fat-saturated T1WI can be used to differentiate hemorrhage from fat because the signal intensity of fat is suppressed [21,22]. Additionally, T1WI is useful for detecting bone metastasis because it is homogeneously hypointense, unlike red marrow, which is heterogeneously hypointense.

### 3.3. DWI Findings

Cervical cancer is hyperintense on DWI because tissue diffusion of water molecules is severely restricted [42,43,44,45] (Figure 1). Accordingly, the contrast between early cervical cancer and normal tissue is high, and the tumor has low apparent diffusion coefficient (ADC) values on ADC map images. As a result, advanced cervical cancer is not difficult to detect on DWI and ADC map images. However, it is not uncommon for early cervical cancer to be invisible on DWI or ADC map images because the tumor volume is small. Therefore, poor depiction of cervical cancer on DWI and ADC map images is indicative of a low tumor burden in early cervical cancer. Lymph node metastasis is frequently hyperintense on DWI and has low ADC values on ADC map images. Bone or ovary metastasis can show strong diffusion restriction but is rare in early cervical cancer. Multi b-value DWI can reflect the early characteristics of water molecule diffusion in pathologic tissues because multi-exponential model DWI can provide more information than mono-exponential model DWI [46,47,48]. Furthermore, high *b* values are superior at differentiating benign and malignant tumors compared with low *b* values [49]. 

### 3.4. CEI Findings

Dynamic CEI is useful to identify the enhancement patterns of cervical cancer, which tends to show early wash-in of contrast material on early phase and early wash-out on late phase dynamic CEI [14,29,39]. As the size of the tumor decreases, a biopsy alone can remove a significant amount of the cervical cancer (Figure 2). Therefore, early cervical cancer is frequently invisible on early-phase dynamic CEI. Marginal enhancement on delayed CEI can be seen along the biopsy defect due to post-biopsy inflammation [14,39]. This inflammation is slightly hyperintense on T2WI but does not show diffusion restriction on DWI [29] (Figure 2). Nevertheless, dynamic CEI findings in early cervical cancer are not well known; such results are primarily reported in advanced cervical cancer patients undergoing concurrent chemo-radiation therapy [50,51,52]. If a cervical cancer is invisible on dynamic CEI, this suggests a small residual cervical cancer or no residual tumor. This MRI sequence is useful to determine if enhancement is present in the tumor [21,22]. Subtracting dynamic CEI from unenhanced 3D T1 mDIXON is useful to determine tumor enhancement when the cervical cancer or metastatic lesion is hyper-intense on unenhanced T1WI.

### 3.5. MRI Findings Indicative of Prognosis

Prognostic factors for early cervical cancer include tumor size, lymphovascular invasion, lymph node metastasis, and depth of stromal invasion [53,54,55]. Among these factors, preoperative tumor size can be measured on MRI; the remaining factors are identified postoperatively by pathologic examination. Accordingly, an invisible tumor on post-biopsy MRI indicates that the tumor burden is much lower than that of early cervical cancer, which is visible on post-biopsy MRI. Park et al. reported no parametrial invasion in women with invisible cervical cancer on MRI [14]. If radical hysterectomy is performed in these patients, it can lead to excessive destruction of the parametrial anatomy and various post-operative morbidities even though long-term survival rates are good (Figure 1 and Figure 2). However, prognosis of early cervical cancer according to cervical cancer subtype was not examined in the study of Park and colleagues [19]. There are several other histologic types of cervical cancer with worse prognosis than squamous cell carcinoma [56].

## 4. Post-Conization MRI

Cervical conization is useful to identify macro-invasive cervical cancer as well as tissue confirm (Figure 2). Post-conization MRI can reveal a triangular tissue defect at the vaginal exocervix [14,29,39]. Residual cancer indicates invasive cervical cancer with an original tumor size greater than 5 mm. Small residual tumor of 5 mm or less is invisible or hard to detect on MRI [14,41]. Therefore, no residual tumor on post-conization MRI indicates a much lower tumor burden but not histological negativity [14,29]. Even though many early cervical cancers are not seen on post-conization MRI, there are small residual cancers with a mean size of approximately 5 mm on hysterectomy specimens [14,41]. For this reason, small cancers at the exocervix are frequently invisible on post-conization MR images. However, radiologists strive to optimize MRI protocols and undergo regular MRI education because dynamic CEI or DWI can depict residual cancer that is invisible on T2WI after conization [57].

## 5. Post-Trachelectomy MRI

Trachelectomy is one of the treatment options for early cervical cancer in women who want to preserve their fertility. MRI is useful to precisely estimate tumor size or location and helps gynecologists determine the surgical options [39,58,59]. Pre-trachelectomy MRI typically reveals early cervical cancer with a low tumor burden without invasion or metastasis [39,58,59]. Post-trachelectomy MRI is also important to identify recurrent tumors during follow-up (Figure 3) [58]. Moreover, follow-up MRI in pregnant women undergoing trachelectomy allows precise measurement of residual cervix size, which can predict adverse pregnancy outcomes [60]. A small recurrent tumor that is not visible on CT can be visible on MRI because of different soft tissue contrast [13,27]. Normal MRI findings following trachelectomy include no tumor at the anastomosis between the uterine body and vagina (Figure 3). Radiating parametrial tissue from the anastomosis is another normal MRI finding in such patients. Great care should be taken to identify local recurrence or metastatic lymph nodes on follow-up MRI.

## 6. MRI for Endocervical Cancer

Endocervical cancer is frequently misdiagnosed as early cervical cancer because this tumor tends to be invisible on physical examination (Figure 4). This tumor is likely to be sampled inappropriately because biopsy devices might not reach the tumor center and only reach the periphery. Therefore, endocervical cancer is diagnosed frequently as carcinoma in situ following biopsy. However, because MRI is good at delineating a tumor in the endocervix as well as one in the exocervix, this imaging modality is useful to determine whether there is an endocervical tumor or if the tumor is in an early or advanced stage. Based on these MRI findings, gynecologists should attempt re-biopsy for appropriate tissue sampling (Figure 4). If cancer tissue cannot be sampled appropriately, transvaginal US-guided or transabdominal-guided biopsy should be used to target an endocervical tumor (Figure 4).

## 7. Early Detection of Recurrent Cancer

MRI is useful for early detection of recurrent cervical cancer following surgery or radiation therapy [58,61,62,63,64] (Figure 5). CT and MRI are used frequently as follow-up imaging modalities in such patients. CT has difficulty identifying small recurrent cancer in the vaginal stump or uterine cervix due to poor soft tissue contrast [61,62,63]. In cases where a recurrent tumor projects into the pelvic cavity or is growing into the uterine body, visual assessment rarely allows early detection on CT. For these reasons, regular follow-up with MRI is necessary to observe signal changes at the vaginal stump or uterine cervix [62]. Early detection makes it possible to choose additional local therapies, such as radiation therapy or thermal ablation, before deciding to pursue more aggressive local or systematic treatments, such as pelvic exenteration or chemotherapy (Figure 5). Furthermore, local treatments can be used to treat oligometastasis if it is detected early.

## 8. Lymph Node Metastasis

It is critical to identify pelvic lymph node metastasis in women with early cervical cancer during pre-treatment or follow-up by CT or MRI (Figure 1). A short-axis lymph node diameter greater than 1 cm on CT or MRI is considered positive for metastasis [40]. However, the size criterion leads to a high frequency of false positive or false negative results; MRI sensitivities with a threshold of 1 cm have been reported to range widely from 30–73% [65,66,67]. Several investigations have determined that DWI is useful to detect metastatic lymph nodes as these have lower ADC values than do benign lymph nodes [68,69,70,71]. A recent systematic review and meta-analysis revealed pooled sensitivity and specificity values of DWI for detecting metastatic lymph nodes of 86% and 84%, respectively [72]. In contrast, lymph node metastasis was detected in only 1.2% of women with invisible cervical cancer on MRI [14] (Figure 1). Invisible tumors on MRI are likely to be a useful finding suggestive of minimal lymphadenectomy in women with early cervical cancer [14,29]. PET-CT is used frequently to detect lymph node metastasis in patients with early cervical cancer. However, a recent meta-analysis reported a pooled sensitivity for PET-CT of only 72% [73].

## 9. Invisible Cancer on 3T MRI as an Indicator

An invisible tumor on MRI is a good indicator that minimally invasive surgery can be performed. Park et al. reported that parametrial invasion was absent in patients with this MRI finding [14]. These investigators demonstrated that vaginal invasion, lymphovascular invasion, and lymph node metastasis are very uncommon in patients with cervical cancer. Therefore, radical hysterectomy, vaginectomy, and lymphadenectomy are excessive surgical resections in these cases. Ongoing development of MRI software and hardware allows detection of smaller cervical cancers because of the resulting improvements in MRI resolution and contrast. For the same reasons outlined above, invisible endometrial cancer on MRI suggests a lower tumor volume, and lymph node dissection can be avoided [74].

## 10. New MRI Techniques

Ultrahigh-field MRI at 7T or higher is expected to be introduced for clinical use in the near future [75,76]. These technical advances likely will improve the amount of anatomical and functional information that can be extracted from tumor images. However, one major technical problem with 7T MR scanning is difficultly achieving homogeneous excitation and homogeneous signal reception across a large field of view because a higher tesla increases main field inhomogeneity [77]. Other problems to be solved include tissue heating, inhomogeneity of the flip angle, and receive sensitivity [77].

PET–MRI is a hybrid imaging system that can provide anatomical and functional information in oncologic patients [78]. Theoretically, this imaging modality is superior to PET-CT in terms of tumor imaging as MRI provides better anatomical information than CT because of the former modality’s excellent soft-tissue contrast. PET–MRI is useful for characterizing tumors of the brain, head and neck, breasts, liver, musculoskeletal system, and urogenitals [79]. Kim et al. [80] fused PET and MRI images in 79 patients with cervical cancer for metastatic work-up. Fusion of PET and MRI images helped these investigators detect additional metastatic lymph nodes in three patients who were node-negative on PET/CT. They concluded that image fusion of PET and MRI can improve the detection of lymph node metastases in patients with cervical cancer.

MRI radiomics allows quantitative evaluation of anatomic structures from MR images and can mitigate variance [81]. The premise of this imaging technique is that quantitative MRI-based features can serve as biomarkers for detecting and characterizing cancers [82]. The results of MRI radiomics should be reproducible for such biomarkers to be useful. Therefore, reproducibility is a basic requirement, and the results should remain stable between two scans at stable clinical conditions [82]. Wormald et al. [83] reported that textural features extracted from the ADC map and T2W images are different between high- and low-volume cervical cancers. Subsequently, MRI radiomics has the potential to predict recurrence in low-volume cervical cancers.

## 11. Conclusions

MRI has the potential to aid gynecologists in managing patients with cervical cancer. Pre-operative MRI helps to estimate the size, volume, location, invasion, and metastasis in patients with cervical cancer and to determine whether a tumor is in an early or advanced stage. Post-operative MRI contributes to early detection of recurrent tumors by gynecologists who must then choose additional local or systemic therapy. Still, useful MRI findings enabling minimally invasive surgery are not common except in patients with invisible cervical cancer on MRI. Further investigations are necessary to discover other useful MRI features that can be used as indicators for minimally invasive treatment of early cervical cancer. As MRI continues to develop, we expect to see improvements in patient selection and treatment outcomes.

## Figures and Tables

**Figure 1 cancers-13-04078-f001:**
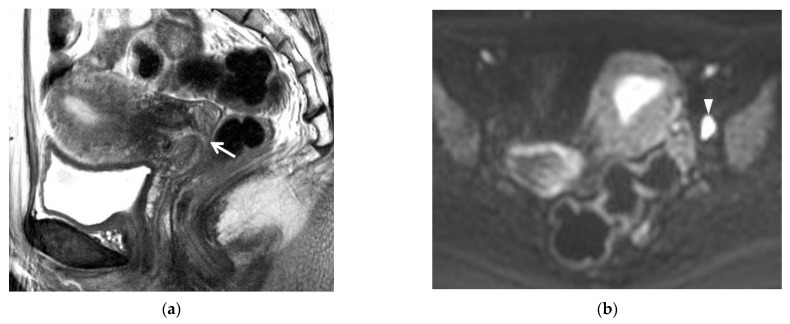
A 45-year-old woman with IB1 cervical cancer: (**a**) the IB1 cervical cancer in the patient undergoing a cone biopsy was not visible on the T2-weighted sagittal image. The arrow indicates the external meatus of the exocervix. (**b**) Diffusion-weighted imaging showed a lymph node (arrowhead) with strong diffusion restriction. No focal lesion was seen in the uterus. Lymph node dissection revealed that the tumor was histologically benign. No parametrial invasion was detected in the radical hysterectomy specimen.

**Figure 2 cancers-13-04078-f002:**
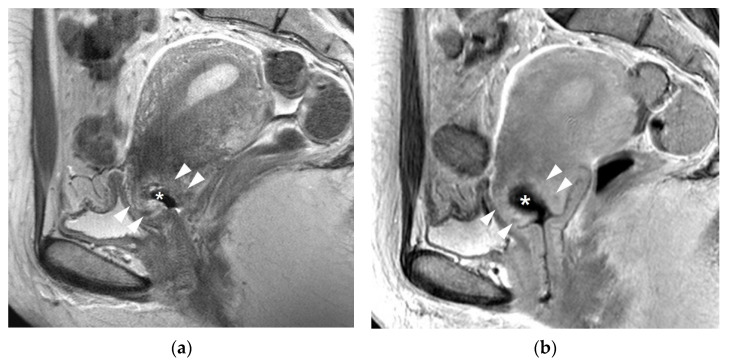
A 42-year-old woman with IA1 cervical cancer: (**a**) T2-weighted sagittal image showed a cone-shape defect (asterisk) after cone biopsy. Arrowheads indicate a poorly demarcated hyperintense lesion indicating post-biopsy inflammation. (**b**). Delayed contrast-enhanced sagittal image shows a hematoma (asterisk) within the cone biopsy defect. Arrowheads indicate a poorly demarcated enhancement indicating post-biopsy inflammation around the tissue defect. Neither residual cancer nor parametrial invasion was detected postoperatively.

**Figure 3 cancers-13-04078-f003:**
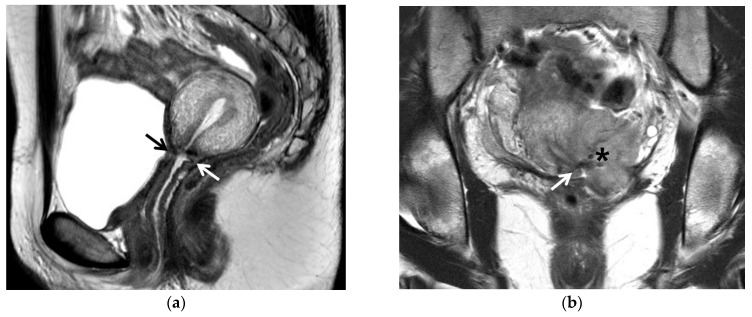
A 30-year-old woman with IB1 cervical cancer: (**a**) T2-weighted sagittal image showed no residual tumor at the trachelectomy site (arrows) in which the uterine body was anastomosed with the vagina. (**b**) T2-weighted coronal image obtained 5 months after trachelectomy showed a recurrent tumor (asterisk) at the left-side margin of the trachelectomy (arrow).

**Figure 4 cancers-13-04078-f004:**
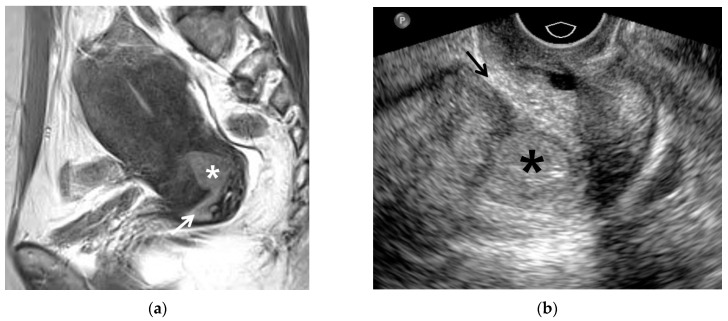
A 53-year-old woman with IB1 endocervical cancer: (**a**) T2-weighted sagittal image clearly depicted an endocervical cancer (asterisk) not detected on visual assessment. The arrow indicates the external meatus of exocervix. This patient had persistent vaginal bleeding prior to admission. (**b**) Longitudinal transvaginal ultrasound image shows a hypoechoic endocervical cancer (asterisk) prior to biopsy. The arrow indicates the external meatus of the exocervix. Histologic diagnosis was adenocarcinoma.

**Figure 5 cancers-13-04078-f005:**
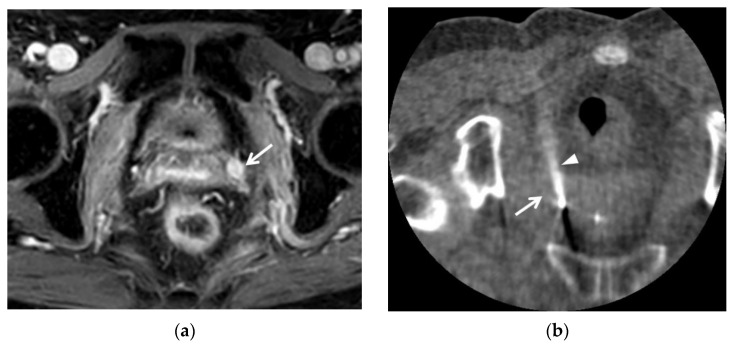
A 62-year-old woman with a recurrent cervical cancer: (**a**) contrast-enhanced T1-weighted axial image showed a small recurrent cancer (arrow) with strong enhancement. This patient underwent concurrent chemo-radiation therapy due to advanced cervical cancer. This tumor was detected early on this follow-up MRI. (**b**) Unenhanced CT axial image showed the recurrent tumor (arrow) in the patient who was prone on the CT table. The arrowhead indicates the radiofrequency electrode (arrowhead) targeting the recurrent cervical cancer perpendicularly. The tumor was completely ablated, and the patient remained free of recurrent tumor for 16 months after radiofrequency ablation treatment.

**Table 1 cancers-13-04078-t001:** Samsung Medical Center parameters for 3T MRI of cervical cancer.

MRI Sequences	T2WI	T1WI	DWI	Dynamic CEI	Delayed CEI
Scan techniques	Fast spin echo	Fast spin echo	Echo planar imaging	3D T1 mDIXON	Fast spin echo
Imaging planes	Axial/sagittal/coronal	Axial	Axial	Axial	Sagittal
Repetition time (ms)	3000–4044	500	6900	3.5	596
Echo time (ms)	100	10	64	0	10
Matrix size	640 × 640	384 × 381	144 × 206	208 × 207	384 × 380
Slice thickness (mm)	4	5	4	4	4
Slice spacing (mm)	4.4	7	4.4	2	4.4
Field of view (cm)	28 × 28 or 30 × 30	28 × 28	28 × 28	28 × 24	30 × 30
Echo train length	16	4	83	2	5
Number of excitations	1	1	3	1	1
B value (s/mm^2^)	NA	NA	0, 100, 1000	NA	NA
Flip angle (°)	90	90	90	10	90

Note—T2WI, T2-weighted image; T1WI, T1-weighted image; DWI, diffusion-weighted image; DCEI, dynamic contrast-enhanced image; NA, not applicable; 3D, three dimensional.

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
