# Peer review of "Useful MRI Findings for Minimally Invasive Surgery for Early Cervical Cancer"

_cancers, 2021, doi:10.3390/cancers13164078_

Round 1

Reviewer 1 Report

Simple summary-

Recommend flipping the order of the simple summary. Start with stating that rad hyst may be over treatment, then go one to say how MRI may help decide who needs surgery.

Line 21- consider "accepts imaging such as MRI as part of the FIGO 2018 updated staging". I do not believe that imaging is a "necessity". 

Line 29- clarify what "minimally invasive means". I think the authors are trying to say "Not radical" but minimally invasive usually means laparoscopic or robotic in this context. I would also define "early" cervical cancer. Stage IA? Stage IB? Early by exam? 

Line 48- change to "quality of life might be negatively affected" 

Line 56- PET scan should also be mentioned 

The purpose of this paper is not clearly stated in the introduction. Was the goal to optimize MRI technique for cervical cancers before treatment? After treatment? 

I found section 3.5 particularly confusing. "MRI invisible" is not a phrase I am familiar with--is it commonly used in radiology literature? Microscopic disease might be a preferred phrase. 

Line 301- the word biomarker is used incorrectly. Change to indicator. 

Overall, there are grammatical issues and sentence structure problems that need to be addressed. It is difficult to understand the purpose of the paper. If it's mean to be a general overview, that is reasonable, but then no conclusions can be drawn. (For example, how can the authors state that MRI is helpful for pre-treatment surgical planning when surgical and survival outcomes are not reported? And there is no information on patient selection? Or how many patients underwent MRI?). 

Author Response

Authors’ responses to reviewer #1 comments

Recommend flipping the order of the simple summary. Start with stating that rad hyst may be over treatment, then go one to say how MRI may help decide who needs surgery.

Response: Thank you for your comments. We agree with you. We will flip the order of the simple summary.

Line 21- consider "accepts imaging such as MRI as part of the FIGO 2018 updated staging". I do not believe that imaging is a "necessity".

Response: Thank you for your comments. We agree with you. We will rephrase “necessity” as "accepts imaging such as MRI as part of the FIGO 2018 updated staging".

Line 29- clarify what "minimally invasive means". I think the authors are trying to say "Not radical" but minimally invasive usually means laparoscopic or robotic in this context. I would also define "early" cervical cancer. Stage IA? Stage IB? Early by exam?

Response: Thank you for your comments. We agree with you. Therefore, we will add the definition in the introduction. It includes all surgical treatments without using a term “radical”. We will also define “early cervical cancer” as stage IA and IB (less than 2 cm).

Line 48- change to "quality of life might be negatively affected"

Response: Thank you for your comments. We agree with you. We will rephrase it to "quality of life might be negatively affected"

Line 56- PET scan should also be mentioned.

Response: Thank you for your comments. We agree with you. We will add “PET scan”.

The purpose of this paper is not clearly stated in the introduction. Was the goal to optimize MRI technique for cervical cancers before treatment? After treatment?

Response: Thank you for your comments. We agree with you. We will revise our purpose to be clearer as follows.

Therefore, the purpose of this review was to show useful MRI features for managing women with early cervical cancer before and after treatment.

I found section 3.5 particularly confusing. "MRI invisible" is not a phrase I am familiar with--is it commonly used in radiology literature? Microscopic disease might be a preferred phrase.

Response: Thank you for your comments. We agree with you. We will rephrase “MRI-invisible cervical cancer” into “invisible cervical cancer on MRI”

Line 301- the word biomarker is used incorrectly. Change to indicator.

Response: Thank you for your comments. We agree with you. We will rephrase “biomarker” into “indicator”.

Overall, there are grammatical issues and sentence structure problems that need to be addressed. It is difficult to understand the purpose of the paper. If it's mean to be a general overview, that is reasonable, but then no conclusions can be drawn. (For example, how can the authors state that MRI is helpful for pre-treatment surgical planning when surgical and survival outcomes are not reported? And there is no information on patient selection? Or how many patients underwent MRI?).

Response: Thank you for your comments. We will ask our manuscript for English editing service. The purpose of our review was to show useful MRI findings for managing patients with early cervical cancers before and after treatment. We have suggested MRI findings to estimate the tumor volume preoperatively. These findings help gynecologists to determine if minimally invasive treatment is possible. However, there are only several retrospective studies demonstrating the utility of MRI for managing early cervical cancer. Further investigations are necessary to assess MRI about how to triage patients and how to contribute to treatment outcomes. We will rephrase our conclusion in the text.

Reviewer 2 Report

You have done an extensive job in gathering all this information. However, in my opinion, this manuscript is more suited as a textbook chapter than a review. Rather than just telling the best approach, you should report different approaches and compare them against each other. I would also prefer that you focus more on how MRI can improve clinical decisions. You should also bring in other approaching techniques to reduce unnecessary lymphadenectomy (LA), such as sentinel node (SN) and FDG-PET-CT or PET-MRI. What you say about LA being used as standard treatment for all early-stage tumors is not true for all institutions/countries. We do not perform LA on invisible tumor on MRI if SN is negative. What about how extensive the LA should be? When to dissect para-aortal lymph nodes and when to just dissect the int/ex/com iliaca and obturator? Which patients should be offered laparoscopic surgery? How can MRI assist in these decisions?

Author Response

Authors’ responses to reviewer #2 comments

You have done an extensive job in gathering all this information. However, in my opinion, this manuscript is more suited as a textbook chapter than a review. Rather than just telling the best approach, you should report different approaches and compare them against each other. I would also prefer that you focus more on how MRI can improve clinical decisions.

Response: Thank you for your comments. I understand what you mean. Initially, I suggested my manuscript as a review article and cancers editors approved it already. The utility of MRI has risen because the new FIGO system accepts the necessity for work up of cancer staging. Accordingly, there are only a few studies demonstrating on how MRI can improve management of early cervical cancer. Despite these limitations, MRI can provide useful Information on clinical decision.

First, if postbiopsy MRI does not show any cervical tumor, it does not indicate radical hysterectomy or lymphadenectomy in stage IB1 cervical cancer. Therefore, this useful MRI finding reduce the risk of postoperative morbidities such as bladder dysfunction, sexual dissatisfaction, anorectal motility disorder, and lymphedema. The quality of life becomes poor in patients with early cervical cancer.

Second, MRI shows the more precise estimation of tumor volume compared to physical examination. Even though an endocervical cancer is clinically staged IA, it can be re-staged as IB or higher on MRI. Moreover, exact measurement of tumor volume is essential to determine if trachelectomy is possible in women who want to have fertility.

Third, MRI is superior to physical examination in terms of parametrial invasion or metastasis. Therefore, even though cervical cancer is staged IB1, it can show parametrial invasion on MRI, leading to changing management into non-surgical treatment. Visual assessment and pelvic palpation alone can barely detect lymph node metastasis.

Fourth, MRI helps to avoid several invasive diagnostic procedures such as intravenous urography, cystoscopy, and sigmoidoscopy. Because of excellent soft tissue contrast, if MRI clear show that cervical cancer invades or does not invade ureter, bladder, and recto-sigmoid colon, intravenous urography, cystoscopy, and sigmoidoscropy are not necessary except cases in which histologic diagnosis should be confirmed.

This utility of MRI influencing the clinical decision has been already stated in the manuscript. Most of all, we have suggested the protocols of 3T MRI. Currently, we have no standard protocols of high tesla MRI in the mangement of cervical cancer. Therefore, our review article helps not only gynecologists to decide their management, but also radiologists to understand how to perform 3T MRI.

You should also bring in other approaching techniques to reduce unnecessary lymphadenectomy (LA), such as sentinel node (SN) and FDG-PET-CT or PET-MRI. What you say about LA being used as standard treatment for all early-stage tumors is not true for all institutions/countries. We do not perform LA on invisible tumor on MRI if SN is negative. What about how extensive the LA should be? When to dissect para-aortal lymph nodes and when to just dissect the int/ex/com iliaca and obturator? Which patients should be offered laparoscopic surgery? How can MRI assist in these decisions?

Response: Thank you for your comments. I understand what you mean. We will revise early cervical cancer into IB1 cervical cancer in which lymph node dissection is mentioned. Please know that the purpose of our review was to report various MRI features for gynecologists to decide management. Invisible cervical cancer on MRI strongly indicates that lymph node dissection is unnecessary in stage IA or that just sampling or no dissection is necessary in 1B1 cancer. Otherwise, 3T MRI does not exclude the possibility of micro-metastasis to lymph node in 1B1 cervical cancer. Still, 3T MRI alone has some limitation in determining if lymph dissection is necessary. Detecting sentinel node metastasis is important to determine if extensive dissection of pelvic and/or para-aortic lymph nodes. Several studies have reported the utility of PET-CT or PET-MRI in detecting sentinel node metastasis. However, they are retrospective studies and their results are not so satisfactory for preoperative detection. As MRI develops continuously, we expect that further investigations using MRI will show more advanced progress in managing early cervical cancer.

Round 2

Reviewer 1 Report

Much closer the publishable, but still needs a good proofreader. I am concerned that the authors make blanket statements about the use of robotic radical hysterectomy (lines 62-64, for example), which has largely been abandoned due the LACC trial (NEJM, 2018, Pedro Ramirez). The majority of practicing gyn/oncs would not perform a minimally invasive surgery, even for "invisible tumor" on MRI. Cold knife cone may be appropriate. I would suggest the authors speak about "too radical a surgery" being performed rather than bringing minimally invasive vs open surgery into the discussion. It's distracting and does not add to their paper.

Author Response

Authors’ responses to Reviewer #1 comments

Much closer the publishable, but still needs a good proofreader. I am concerned that the authors make blanket statements about the use of robotic radical hysterectomy (lines 62-64, for example), which has largely been abandoned due the LACC trial (NEJM, 2018, Pedro Ramirez). The majority of practicing gyn/oncs would not perform a minimally invasive surgery, even for "invisible tumor" on MRI. Cold knife cone may be appropriate. I would suggest the authors speak about "too radical a surgery" being performed rather than bringing minimally invasive vs open surgery into the discussion. It's distracting and does not add to their paper.

Response: Thank you for your comments. We understand what you mean. Our review article is focused on utility of 3T MRI in considering preoperative management. This imaging modality has provided better resolution and faster scan time compared to 1.5T MRI. However, there is no established consensus on patient preparations, imaging protocols, and imaging features in performing 3T MRI for gynecologic oncologic imaging. For this reason, many institutes do not show good image quality even though they use 3T MRI. Our review states how to prepare for 3T MRI scan, what sequences are necessary, and what imaging features are suggesting low tumor volume. Many leading papers about 3T MRI have been published on our hospital, Samsung Medical Center. We can depict detailed morphology of cervical cancer as 3T MRI is upgraded. Initially, our gynecologists did not believe the clinical significance of invisible cervical cancer on 3T MRI. Currently, they get to know that this MRI is a strong indicator to low tumor volume. Accordingly, they try to minimize parametrectomy and lymph node dissection in invisible 1B1 cervical cancer on MRI. As a result, higher strength MRI will show more precise tumor extent, so that the MRI findings become identical to histologic findings. Several papers have already demonstrated that invisible 1B1 cervical cancers on 3T MRI have no residual cancer in almost half of the cases. Moreover, none of invisible 1B1 cancers invaded parametrium even though residual cervical cancers were detected on radical hysterectomy specimen. Lymph node metastasis was extremely low in these cases. Excessive parametrectomy and lymph node dissection were so unnecessary surgical procedures that they could not avoid post-operative morbidities such as voiding difficulty, sexual dysfunction, anorectal disorder, and lymph edema.

Another purpose of our review is to help radiologists know how to make good patient preparations, to make good gynecologic MR images at 3T scanner, and what imaging features are important for surgical plan. These issues are of great importance to radiologists who should read 3T MR images in patients with cervical cancer. If they become familiar with patient preparations, MRI sequences, and imaging features at 3T, they will help gynecologists to determine surgical extent in early cervical cancer and be already prepared for higher strength MRI. We will add these statements in the manuscript.

Reviewer 2 Report

General comments: This review is comprehensive and informative. The language and grammar have been improved during review. Still, the manuscript would benefit from having an extended part explaining the purpose and the outline. I would prefer that you insert that at the end of the introduction or as the first paragraph of section 2. You need to explain why you on several occasions present your protocols as the default/gold standard. I do not understand the rationale behind that. I review is supposed to be objective. The text is very repetitive around the different risks for visible versus invisible tumors. Could you shorten the text and make sure you do not repeat yourself unnecessary?

Specific comments:

Simple Summary: The first sentence should be rephrased. I think what you mean is that radical hysterectomy and lymph node dissection is extensive procedures with severe post-operative morbidities and should be avoided on patients with low risk of recurrence. Still, due to lack of good prognostic tools, radical surgery is performed on most early-stage patients with visible tumor (FIGO 1B1(?)) leading to overtreatment and unnecessary morbidities.

Introduction:

Line 37-38: This statement is wrong. HPV vaccines will have little or no effect on the prevalence of cervical cancer the next 15-20 years. The HPV vaccine Gardasil was approved in 2006/2007 and Cervarix in 2009. It took a few years before it was introduced in the national schedule and currently 100 countries covering only 30% of the global population has introduced the vaccine. Consequently, the first girls that were vaccinated from 2006 and onwards are currently no more than 25-30 years. They are too young to have any substantial impact on the prevalence of cervical cancer – a disease with a median age of around 45. This effect will not be seen until 15-20 years from now. Therefore, screening is still the intervention factor that affects the prevalence of HPV-related cervical cancer, not vaccination.

Line 38-42: In the next sentences, you need to simplify/shorten down. The point is that screening leads to early detection. Hence, with higher screening rates, a higher portion of the detected cases will be early stage.

Line 44-45: Could oncologic outcomes be rephrased as clinical outcomes, or survival? These are more common phrases.

Line 49: This sentence seems to be grammatically wrong. What do you refer to as ‘it’ here?

Line 269: There are several other histologic types of cervical cancer with worse prognosis than gastric-type adenocarcinoma (PMID: 28599900).

Author Response

Authors responses to reviewer #2 comments

General comments: This review is comprehensive and informative. The language and grammar have been improved during review. Still, the manuscript would benefit from having an extended part explaining the purpose and the outline. I would prefer that you insert that at the end of the introduction or as the first paragraph of section 2. You need to explain why you on several occasions present your protocols as the default/gold standard. I do not understand the rationale behind that. I review is supposed to be objective. The text is very repetitive around the different risks for visible versus invisible tumors. Could you shorten the text and make sure you do not repeat yourself unnecessary?

Response: Thank you for your comments. The purpose of our review was to help radiologists and gynecologists with patient preparation, 3T MRI protocols, and imaging features. Therefore, we will add another purpose at the end of introduction. But also, we will minimize use of the visible versus invisible tumors.

Specific comments:

Simple Summary: The first sentence should be rephrased. I think what you mean is that radical hysterectomy and lymph node dissection is extensive procedures with severe post-operative morbidities and should be avoided on patients with low risk of recurrence. Still, due to lack of good prognostic tools, radical surgery is performed on most early-stage patients with visible tumor (FIGO 1B1(?)) leading to overtreatment and unnecessary morbidities.

Response: Thank you for your comments. We will rephrase the first sentence with your comments.

Introduction:

Line 37-38: This statement is wrong. HPV vaccines will have little or no effect on the prevalence of cervical cancer the next 15-20 years. The HPV vaccine Gardasil was approved in 2006/2007 and Cervarix in 2009. It took a few years before it was introduced in the national schedule and currently 100 countries covering only 30% of the global population has introduced the vaccine. Consequently, the first girls that were vaccinated from 2006 and onwards are currently no more than 25-30 years. They are too young to have any substantial impact on the prevalence of cervical cancer – a disease with a median age of around 45. This effect will not be seen until 15-20 years from now. Therefore, screening is still the intervention factor that affects the prevalence of HPV-related cervical cancer, not vaccination.

Response: Thank you for your comments. We will rephrase the line 37-38 with your comments.

Line 38-42: In the next sentences, you need to simplify/shorten down. The point is that screening leads to early detection. Hence, with higher screening rates, a higher portion of the detected cases will be early stage.

Response: Thank you for your comments. We agree with you. We will rephrase the line 38 – 42 with your comments.

Line 44-45: Could oncologic outcomes be rephrased as clinical outcomes, or survival? These are more common phrases.

Response: Thank you for your comments. We agree with you. We will rephrase oncologic outcomes with long-term survival.

Line 49: This sentence seems to be grammatically wrong. What do you refer to as ‘it’ here?

Response: Thank you for your comments. We will rephrase the line 49 as follows. Minimally invasive treatment is necessary to reduce such functional disability and is defined as all surgical treatments without using a term “radical”.

Line 269: There are several other histologic types of cervical cancer with worse prognosis than gastric-type adenocarcinoma (PMID: 28599900).

Response: Thank you for your comments. We will add your comments.

Round 3

Reviewer 2 Report

Simple summary: Thank you for taking my suggestion into consideration. I think it reveals the main message of this review in a better now. Still, I think you have to insert ‘most’ or ‘the majority of’ before patients with early staged cervical cancer to pinpoint what you mean.

Line 37-38: The text should be improved. I realize that you cannot write everything that I presented within the manuscript text, but you should explain enough to make the sentence make sense. Maybe you should just drop mentioning this about vaccines? You could say something about the prevalence and death rates instead:

More than 600 000 women are diagnosed with cervical cancer annually and the disease causes over 300 000 deaths worldwide. You should use this reference: https://acsjournals.onlinelibrary.wiley.com/doi/10.3322/caac.21660

However, due to the establishment of worldwide screening programs leading to early detection of disease, an increasing potion of newly detected cases are early stage.

Section 3.5: The inserted sentence does not fit with the previous sentence. My comment was on why you only mention gastric-type adenocarcinoma as having worse prognosis. In fact, histologic type is an important prognostic factor, so maybe you should put it in the first sentence of the paragraph instead? If Park et al had several gastric type adenocarcinomas in their study, you should say so. If not, you can simply say that they did not account for histology and drop the sentence where you mention gastric-type adenocarcinoma.

Author Response

Authors’ responses to reviewer #2 comments.

Simple summary: Thank you for taking my suggestion into consideration. I think it reveals the main message of this review in a better now. Still, I think you have to insert ‘most’ or ‘the majority of’ before patients with early staged cervical cancer to pinpoint what you mean.

Response: Thank you for your comments. We will insert it.

Line 37-38: The text should be improved. I realize that you cannot write everything that I presented within the manuscript text, but you should explain enough to make the sentence make sense. Maybe you should just drop mentioning this about vaccines? You could say something about the prevalence and death rates instead:

More than 600 000 women are diagnosed with cervical cancer annually and the disease causes over 300 000 deaths worldwide. You should use this reference: https://acsjournals.onlinelibrary.wiley.com/doi/10.3322/caac.21660

However, due to the establishment of worldwide screening programs leading to early detection of disease, an increasing potion of newly detected cases are early stage.

Response: Thank you for your comments. We will rephrase the line 37-38 with your comments.

Section 3.5: The inserted sentence does not fit with the previous sentence. My comment was on why you only mention gastric-type adenocarcinoma as having worse prognosis. In fact, histologic type is an important prognostic factor, so maybe you should put it in the first sentence of the paragraph instead? If Park et al had several gastric type adenocarcinomas in their study, you should say so. If not, you can simply say that they did not account for histology and drop the sentence where you mention gastric-type adenocarcinoma.

Response: Thank you for your comments. We understand what you mean. The sentence will be deleted.
